# Immune status changing helps diagnose osteoarticular tuberculosis

Tuo Liang[1], Jiarui Chen[1], GuoYong Xu[1], Zide Zhang[1], Jiang Xue[1], Haopeng Zeng[1], Jie Jiang[1], Tianyou Chen[1], Zhaojie Qin[1], Hao Li[1], Zhen Ye[1], Yunfeng Nie[2], Chong Liu[1], Xinli Zhan[1]*

1 Spine and Osteopathy Ward, The First Affiliated Hospital of Guangxi Medical University, Nanning, Guangxi, People's Republic of China, 2 Guangxi Medical University, Nanning, Guangxi, People's Republic of China

* zhanxinli@stu.gxmu.edu.cn

## Abstract

### Objective

This study is aimed to develop a new nomogram for the clinical diagnosis of osteoarticular tuberculosis (TB).

### Methods

xCell score estimation to obtained the immune cell type abundance scores. We downloaded the expression profile of GSE83456 from GEO and proceed xCell score estimation. The routine blood examinations of 326 patients were collected for further validation. We analyzed univariate and multivariate logistic regression to identified independent predicted factor for developing the nomogram. The performance of the nomogram was assessed using the receiver operating characteristic (ROC) curves. The correlation of ESR with lymphocytes, monocytes, and ML ratio was performed and visualized in osteoarticular TB patients.

### Results

Compared with the healthy control group in the dataset GSE83456, the xCell score of basophils, monocytes, neutrophils, and platelets was higher, while lymphoid was lower in the EPTB group. The clinical data showed that the cell count of monocytes were much higher, while the cell counts of lymphocytes were lower in the osteoarticular TB group. AUCs of the nomogram was 0.798 for the dataset GSE83456, and 0.737 for the clinical data. We identified the ML ratio, BMI, and ESR as the independent predictive factors for osteoarticular TB diagnosis and constructed a nomogram for the clinical diagnosis of osteoarticular TB. AUCs of this nomogram was 0.843.

### Conclusions

We demonstrated a significant change between the ML ratio of the EPTB and non-TB patients. Moreover, we constructed a nomogram for the clinical diagnosis of the osteoarticular TB diagnosis, which works satisfactorily.

**Data Availability Statement:** The GEO database dataset (GEO ID: GSE83456) generated and/or analyzed during the current study are available in the GEO database (www.ncbi.nlm.nih.gov/gds).

**Funding:** This work was sponsored by the National Natural Science Foundation of China (81560359); National Natural Science Foundation of China (81860393). The funders had no role in study design, data collection and analysis, decision to publish, or preparation of the manuscript.

**Competing interests:** The authors have declared that no competing interests exist.

**Abbreviations:** TB, Tuberculosis; EPTB, Extrapulmonary tuberculosis; TME, Tumor microenvironment; TIICs, Tumor-infiltrating immune cell; GSEA, Gene Set Enrichment Analysis; GO, gene ontology; KEGG, Kyoto Encyclopedia of Genes and Genomes; DEGs, differentially expressed genes; ROC, receiver operating characteristic; ESR, erythrocyte sedimentation rate; BMI, body mass index.

## Introduction

World health organization (WHO) estimates that 1.8 billion people, about one-fourth of the global population, are *Mycobacterium tuberculosis (M.tb)* infected, out of which roughly 10 million contracted tuberculosis (TB) and 1.6 million died of the disease [1–3]. Therefore, TB remains a leading global public health problem [4, 5]. Till now, TB has been described in all virtual tissues or organs, including the spine, lymph nodes, abdomen, genitourinary tract, skin, joints, and meningeal [6, 7]. Extrapulmonary tuberculosis (EPTB) incidents are higher in immune-compromised individuals and represents 15% of the global TB cases [8]. However, an individual's immunity status decides whether patients with *M.tb* would develop active TB or not [9].

Despite the existence of guidelines for the diagnosis and treatment of TB [10], there tend to be cases of EPTB that present with atypical manifestations such as local pain, weight loss, night sweat and fever, which make the diagnosis difficult [11]. In addition, the anatomical sites frequently involves in EPTB are not easily accessible and require invasive procedures for diagnostic confirmation. The availability of little information is the reason for the difficulty and delay in the diagnosis of EPTB. Therefore, an urgent need arises to invent new methods for EPTB diagnosis.

The absolute number of monocytes or lymphocytes in peripheral blood or yet the ratio of monocytes to lymphocytes (ML ratio) has prognostic value in diseases such as hematopathy and tumors [12]. Previous studies have demonstrated that the ML ratio shows a predictive value for active TB [13]. The development of osteoarticular TB is the result of an immune system disorder. However, spine TB is the most common kind of osteoarticular TB, which influences the proportion of immune cells. Therefore, the discovery of immune status associated with osteoarticular TB can advance osteoarticular TB diagnosis.

In this study, the expression profile of dataset GSE83456 was downloaded to obtain the xCell score of the cell type, including lymphocytes, monocytes, neutrophils, eosinophils, basophils, erythrocyte, and platelet that belongs to the blood routine examination items. We found that the immune status of individuals changes during TB infection. ML ratio was found significantly up-regulated in the EPTB group than that in the non-TB group. Based on this result, we constructed a novel nomogram, which incorporated easy access to clinical characteristics like ML raito, erythrocyte sedimentation rate (ESR) and body mass index (BMI), for osteoarticular TB diagnosis.

## Patients and methods

### Microarray data and xCell score estimation

Gene Expression Omnibus (GEO, https://www.ncbi.nlm.nih.gov/geo/) is an open database from where we downloaded the gene expression profiles. The GSE83456 with platform GPL10558 (Illumina HumanHT-12 V4.0 expression beadchip) was downloaded and used to extracted blood sample profiles of 47 EPTB and 61 healthy controls (HCs) from the total 202 samples for further analysis. We analyzed the xCell score using the R package 'xCell' (https://github.com/dviraran/xCell), which allowed us to obtained 64 immune cell type abundance scores [14]. Next, the xCell score of lymphocytes, monocytes, neutrophils, eosinophils, basophils, erythrocyte, and platelet belonging to the blood routine examination items was picked out. The xCell score of lymphocytes was a constitution of 21 subtypes (S1 Fig) of lymphoid cells (B cells, T cells and NK cells). This score was consistent with the method of lymphocyte count in the routine blood examination.

## Patients

Subjects volunteering for the study had signed informed consent forms. In addition, the Ethics Committee of The First Affiliated Hospital of Guangxi Medical University approved this study.

From 2012 to 2018, we consecutively screened out 173 osteoarticular TB patients from The First Affiliated Hospital of Guangxi Medical University. Following were the included diagnostic criteria for osteoarticular TB: (1) patients with typical symptoms of tuberculous infection, including low fever, night sweats, weight loss, and fatigue; (2) patients with positive *M.tb* antibody; (3) patients with spinal cord compression symptoms, such as pain, myodynamia, muscle tension, tendon reflexes, limited activity, and spinal deformities; (4) patients with the typical features of spinal TB on imageology, such as bone marrow edema, endplate erosion, vertebral destruction, and spinal compression; and (5) patients with tuberculous granuloma [15]. Exclusion criteria were set for osteoarticular TB patients: (1) co-infection with other types of bacteria, virus, and co-morbidities; (2) co-occurrence with tumor (3) already receiving anti-tuberculosis drug treatment; and (4) recurrent tuberculosis.

In The First Affiliated Hospital of Guangxi Medical University, from 2012 and 2018, we randomly screened out the non-TB patients from all the inpatients diagnosed with lumbar disc herniation or lumbar spinal stenosis. After identifying patients, 326 eligible patients were enrolled, including 164 osteoarticular TB patients and 162 non-TB patients.

## Statistical analysis

Using the Student's t-test, we analyzed continuous variables such as xCell score of monocytes, peripheral blood of monocytes counts, or lymphocytes. To identify the independent risk factors in the dataset GSE83456 and the clinical data, we performed the univariate and multivariate logistics regression analyses using the R package 'rms' (https://cran.rstudio.com/bin/windows/contrib/4.0/rms_6.1-0.zip). Also, we constructed and compared the nomograms based on the logistics regression result obtained. The performance of the nomogram were assessed using the receiver operating characteristic (ROC) curves. Furthermore, factors in the clinical cohort sample were evaluated via univariate and multivariate logistic regression. The correlation of ESR with lymphocytes, monocytes, and ML ratio in osteoarticular TB patients was performed and visualized using the R package 'corrplot' (https://cran.rstudio.com/bin/windows/contrib/4.0/corrplot_0.84.zip). The value of $P < 0.05$ was considered a significant difference.

## Gene Set Enrichment Analysis (GSEA)

Using GSEA (4.0.3), we identified the potential biological mechanisms of ML ratio, which were involved in the impact of osteoarticular TB [16]. The gene set permutations with 1000-times were conducted to acquire the normalized enrichment score (NES). The normal P-value $< 0.05$ and false discovery rate (FDR) $< 0.25$ was used to quantify statistically significant enrichment.

## Results

### Patient baseline characteristics

Table 1 illustrates the baseline characteristics collected for the 326 patients, such as age, sex, profession, nationality, BMI, ESR. In this study, spine TB accounted for 93.9% and constituted the majority of osteoarticular TB. BMI and hemoglobin were much lower in the osteoarticular TB group than in the non-TB group. However, ESR was much higher in the osteoarticular TB group (Table 1).

**Table 1. Baseline characteristics of patients.**

| Characteristics | Osteoarticular TB (N = 164) | HC (N = 162) | Overall (N = 326) | P value |
|---|---|---|---|---|
| **Age** | | | | |
| Mean (SD) | 45.2 (17.0) | 48.0 (17.2) | 46.6 (17.2) | 0.137 |
| Median [Min, Max] | 46.5 [3.00, 78.0] | 51.0 [5.00, 86.0] | 49.0 [3.00, 86.0] | |
| **Sex** | | | | |
| Female | 69 (42.1%) | 69 (42.6%) | 138 (42.3%) | 1 |
| Male | 95 (57.9%) | 93 (57.4%) | 188 (57.7%) | |
| **Nationality** | | | | |
| Han | 80 (48.8%) | 97 (59.9%) | 177 (54.3%) | 0.158 |
| Zhuang | 75 (45.7%) | 59 (36.4%) | 134 (41.1%) | |
| Yao | 8 (4.9%) | 4 (2.5%) | 12 (3.7%) | |
| Others | 1 (0.6%) | 2 (1.2%) | 3 (0.9%) | |
| **Profession** | | | | |
| Farmer/Worker | 100 (61.0%) | 86 (53.1%) | 186 (57.1%) | 0.153 |
| Office clerk | 9 (5.5%) | 10 (6.2%) | 19 (5.8%) | |
| Student | 7 (4.3%) | 17 (10.5%) | 24 (7.4%) | |
| Others | 48 (29.3%) | 49 (30.2%) | 97 (29.8%) | |
| **BMI** | | | | |
| Mean (SD) | 20.1 (2.79) | 22.5 (4.17) | 21.3 (3.76) | <0.001 |
| Median [Min, Max] | 19.8 [13.2, 29.4] | 22.7 [13.5, 36.9] | 21.0 [13.2, 36.9] | |
| Missing | 20 (12.2%) | 10 (6.2%) | 30 (9.2%) | |
| **Location** | | | | |
| - | 0 (0%) | 162 (100%) | 162 (49.7%) | <0.001 |
| joint tuberculosis | 10 (6.1%) | 0 (0%) | 10 (3.1%) | |
| spine tuberculosis | 154 (93.9%) | 0 (0%) | 154 (47.2%) | |
| **Hemoglobin** | | | | |
| Mean (SD) | 119 (17.6) | 127 (16.7) | 123 (17.6) | <0.001 |
| Median [Min, Max] | 121 [76.4, 153] | 128 [75.8, 167] | 124 [75.8, 167] | |
| **Erythrocyte** | | | | |
| Mean (SD) | 4.71 (3.01) | 4.60 (0.663) | 4.65 (2.19) | 0.646 |
| Median [Min, Max] | 4.46 [2.83, 42.0] | 4.56 [3.14, 7.07] | 4.51 [2.83, 42.0] | |
| **White blood cell** | | | | |
| Mean (SD) | 7.21 (2.71) | 7.62 (2.57) | 7.41 (2.65) | 0.163 |
| Median [Min, Max] | 6.78 [2.55, 17.2] | 7.08 [3.10, 21.2] | 6.90 [2.55, 21.2] | |
| **Lymphocytes** | | | | |
| Mean (SD) | 1.52 (0.798) | 2.06 (0.780) | 1.79 (0.834) | <0.001 |
| Median [Min, Max] | 1.38 [0.268, 7.21] | 1.95 [0.382, 5.55] | 1.67 [0.268, 7.21] | |
| **Monocytes** | | | | |
| Mean (SD) | 0.645 (0.269) | 0.582 (0.205) | 0.614 (0.241) | 0.017 |
| Median [Min, Max] | 0.589 [0.20, 1.72] | 0.553 [0.04, 1.52] | 0.568 [0.04, 1.72] | |
| **Neutrophil** | | | | |
| Mean (SD) | 4.73 (2.25) | 4.72 (2.49) | 4.72 (2.37) | 0.964 |
| Median [Min, Max] | 4.24 [1.16, 14.7] | 4.07 [1.55, 19.5] | 4.14 [1.16, 19.5] | |
| **Eosinophils** | | | | |
| Mean (SD) | 0.278 (0.474) | 0.229 (0.172) | 0.254 (0.357) | 0.219 |
| Median [Min, Max] | 0.189 [0, 5.73] | 0.201 [0, 0.976] | 0.198 [0, 5.73] | |
| **Basophils** | | | | |

**Table 1.** (Continued)

| Characteristics | Osteoarticular TB | HC | Overall | P value |
|---|---|---|---|---|
| | (N = 164) | (N = 162) | (N = 326) | |
| Mean (SD) | 0.034 (0.021) | 0.036 (0.019) | 0.035 (0.020) | 0.225 |
| Median [Min, Max] | 0.031 [0, 0.162] | 0.033 [0, 0.133] | 0.032 [0, 0.162] | |
| **Platelet** | | | | |
| Mean (SD) | 290 (85.3) | 250 (80.1) | 270 (85.1) | <0.001 |
| Median [Min, Max] | 274 [78.5, 562] | 239 [98.0, 630] | 256 [78.5, 630] | |
| **ESR** | | | | |
| Mean (SD) | 38.6 (23.6) | 19.5 (19.6) | 30.0 (23.8) | <0.001 |
| Median [Min, Max] | 34.0 [1.00, 109] | 11.0 [2.00, 94.0] | 24.0 [1.00, 109] | |
| Missing | 6 (3.7%) | 34 (21.0%) | 40 (12.3%) | |

## Osteoarticular tuberculosis group vs. non-TB group

The xCell score of 64 subtypes for each sample was calculated based on the expression file (S1 File). The xCell score of lymphocytes was sum of all xCell score of lymphoid cells (B cells, T cells and NK cell) in the dataset GSE83456 (S1 Fig). Compared with the HC group in the dataset GSE83456, the EPTB group has a higher xCell score of basophils, monocytes, neutrophils, and platelets, but lower xCell score of lymphocytes (Fig 1A). The cell counts of monocytes and platelets were much higher while that of lymphocytes were lower in the osteoarticular TB group of the clinical data (Fig 1B). Monocytes, lymphocytes and platelet were significantly different between the osteoarticular TB and non-TB group.

## A high ML ratio is associated with osteoarticular TB

The ML ratio in the dataset GSE83456 represents the ratio of xCell score of monocytes to lymphocytes. The ML ratio in the clinical data represents the ratio of cell counts of monocytes to lymphocytes. In the EPTB group, the ML ratio was significantly higher in the dataset GSE83456 (Fig 2A) and clinical data (Fig 2B). We identified the ML ratio as an independent risk factor for the diagnosis of EPTB; therefore, nomograms (S2 Fig) based on the ML ratio were constructed for both dataset GSE83456 and clinical data. ROC analysis showed the AUC of the nomogram was 0.798 for the dataset GSE83456, and 0.737 for the clinical dataset (Fig 2C).

GSEA analysis showed that 22 hallmarks pathways were significantly enriched in the EPTB group, while none enriched in the HC group. Fig 2D shows 12 representative pathways, including apoptosis, reactive oxygen species (ROS), inflammatory response, interferon-α/γ response, complement, IL6/JAK/STAT3, and TNFα via NF-κB. Moreover, GSEA analysis also showed that nine significant enriched pathways in the high ML ratio phenotype were consistent with those of the EPTB group (Fig 2E).

## Nomogram for osteoarticular TB diagnosis

Fig 3A shows a nomogram based on three independent risk factors (ML ratio, ESR, and BMI), which we constructed for advancing clinical diagnosis of osteoarticular TB (Table 2). AUC of this nomogram was 0.843 (Fig 3B); also, the calibration curves indicated a satisfactory agreement between nomogram prediction and actual probabilities (Fig 3C). We further evaluated a correlation between monocytes, lymphocytes, and ESR. Monocytes and ML ratio was positively correlated with ESR while lymphocytes was negatively correlated with ESR. Fig 3D illustrates ML ratio was positively correlated with ESR (cor = 0.439, P < 0.0001).

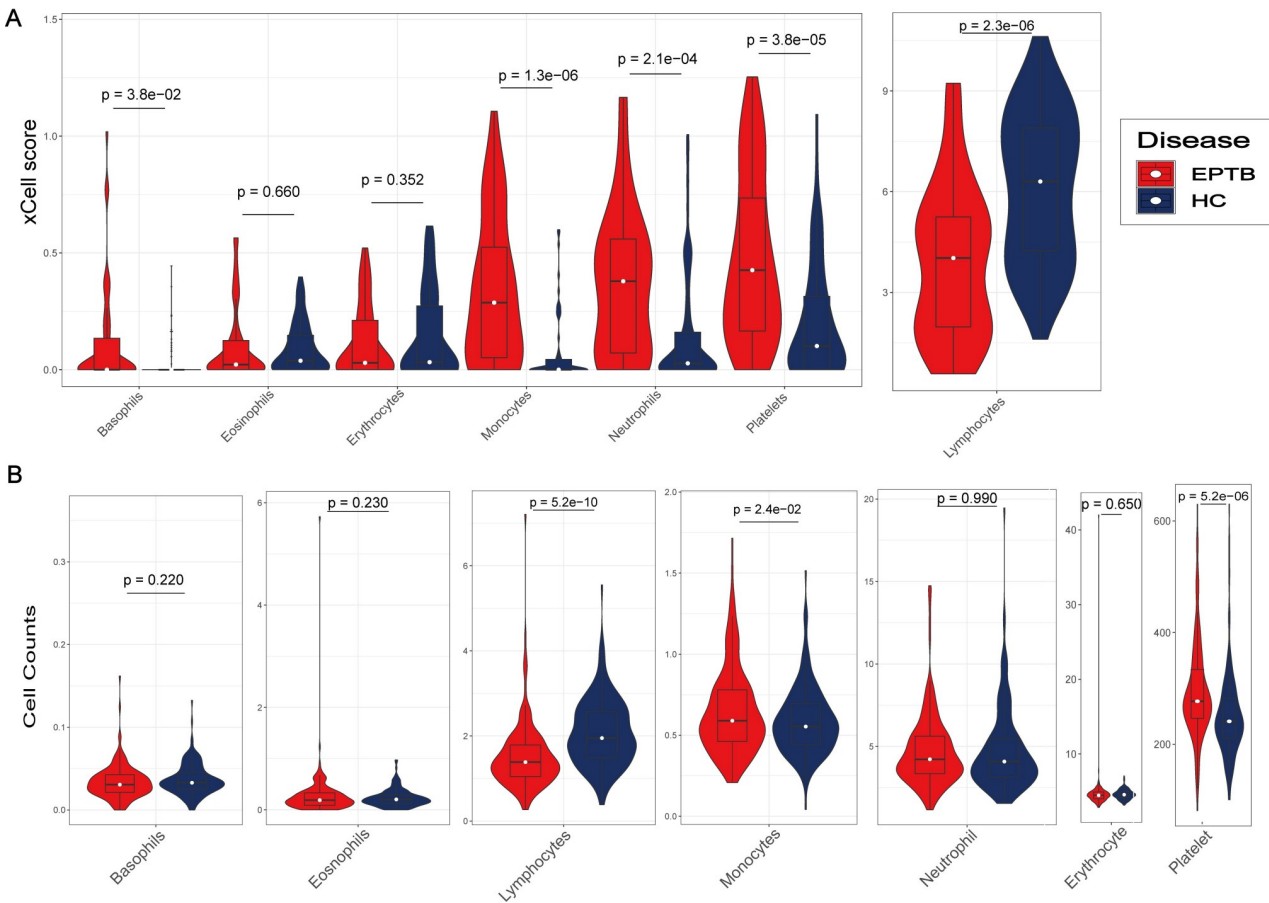

**Fig 1. Different cell type between EPTB and healthy controls.** (A) Violin plot showed the xCell score of 7 kinds of blood cells in dataset GSE83456. (B) Violin plot showed the cell counting of 7 kinds of blood cells in clinical data. (C) Violin plot showed the cell rate of 5 kinds of blood cells in clinical data.

## Discussion

At present, TB poses a global threat for both developing and developed countries [17, 18], the innate immune response representing one of the most critical determinants associated with the outcome of EPTB infection [19]. In this study, the xCell score of monocytes and platelets is significantly higher, while that of lymphocytes is lower in the EPTB group in dataset GSE83456. Moreover, the ML ratio of the xCell score is significantly higher in the EPTB group. GSEA results showed pathways closely related to the inflammation, such as apoptosis, reactive oxygen species (ROS), inflammatory response, and interferon-α/γ response, were significantly enriched. Previous studies suggest that ROS and interferon-α/γ might be associated with *M.tb* immune escape and disease progression in infected humans [20, 21]. Micheliolide and nitric oxide play an anti-inflammatory role in *M.tb* infection by inhibiting NF-κB that is a ubiquitously existed transcription factor family [22, 23]. These results demonstrated *M.tb* infection activates these inflammation-related pathways and plays an significant role in the pathogenesis of EPTB. The pathways significantly enriched in the EPTB group were also enriched in the high ML ratio phenotype, which indicates that a high ML ratio is a crucial characteristic of EPTB and is consistent with a previous study [13].

Understanding the immune responses protecting from infection or progression to disease is crucial to allow the development of diagnostic tools for the efficient prevention and

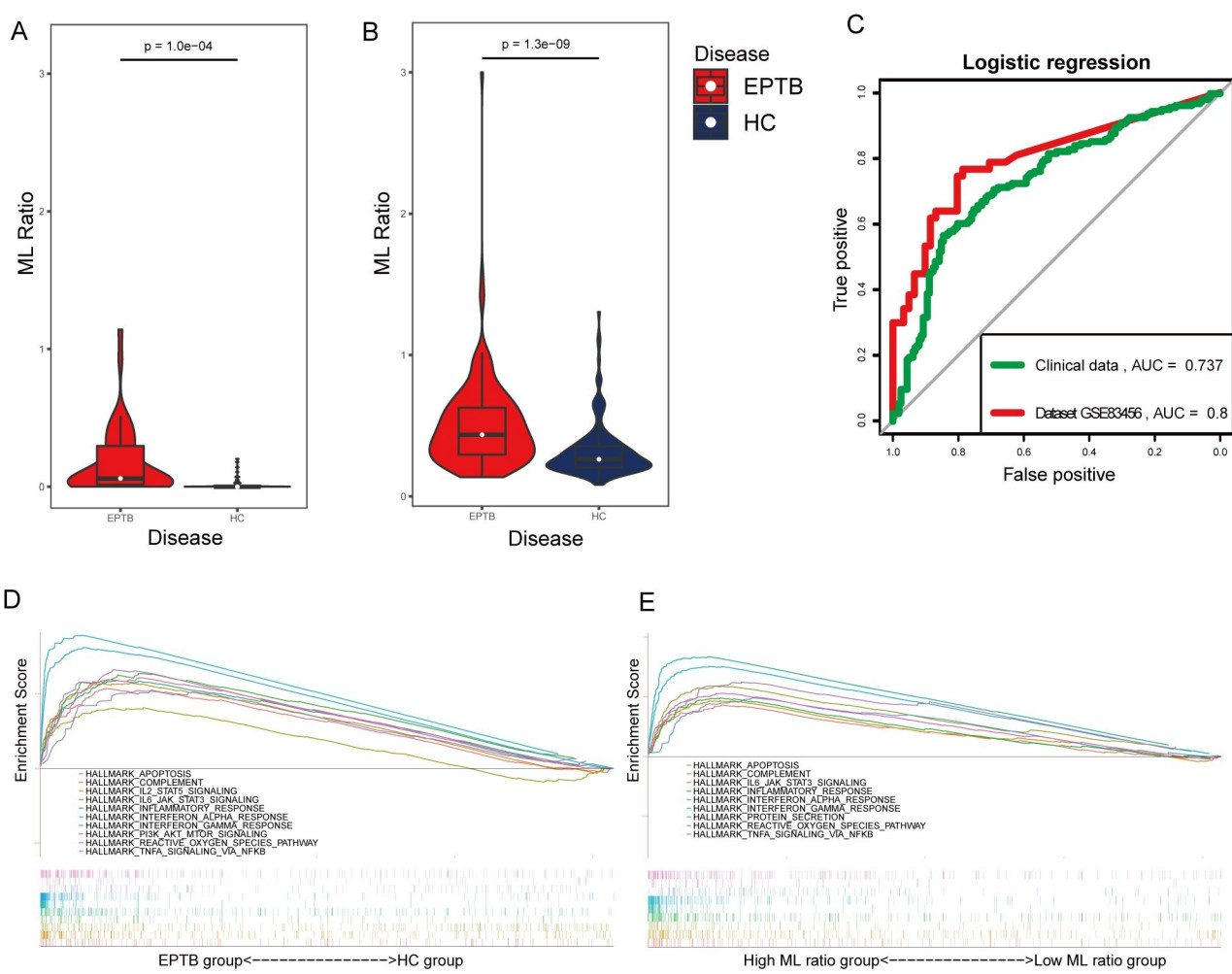

**Fig 2. T test of ML ratio between EPTB and healthy controls and GSEA analysis.** (A) Violin plot showed the ML ratio between EPTB and healthy controls in dataset GSE83456. (B) Violin plot showed the ML ratio between EPTB and healthy controls in clinical data. (C) AUCs of the nomogram based on ML ratio in GSE83456 and clinical data. (D) The enriched gene sets in HALLMARK collection by the EPTB samples. (E) The enriched gene sets in HALLMARK collection by the high ML ratio samples.

management of EPTB. The assumption is further confirmed by the analysis of the clinical data. It had been proved that the cell counts and frequency of monocytes and lymphocytes were significantly different between TB disease and HCs [24], which were consistent with our results. Lymphocytes are thought to be the primary effector cells in TB immunity, while myeloid cells as the primary host cell for infection. Therefore, the relative abundance may reflect a balance between effector and target cells. An alternative explanation is that the relative abundance of these cell types could be a marker of hematopoietic parameters associated with TB. The ratio of myeloid transcripts to lymphoid transcripts were altered in inflammation-associated disease. Altogether, these data support the hypothesis that a high or a low ML ratio may be a correlate of risk for TB. Naranbhai and colleagues demonstrated that the ML ratio may be a readily available tool to identify the risk of TB during HIV infection of patients with acceptable combination of antiretroviral therapy [25, 26].

So far, the osteoarticular TB diagnosis relies on the comprehensive analysis of the history, imaging, and blood test. However, the challenges in the clinical identification of osteoarticular TB are compounded by a diagnostic armamentarium with significant limitations. Based on the

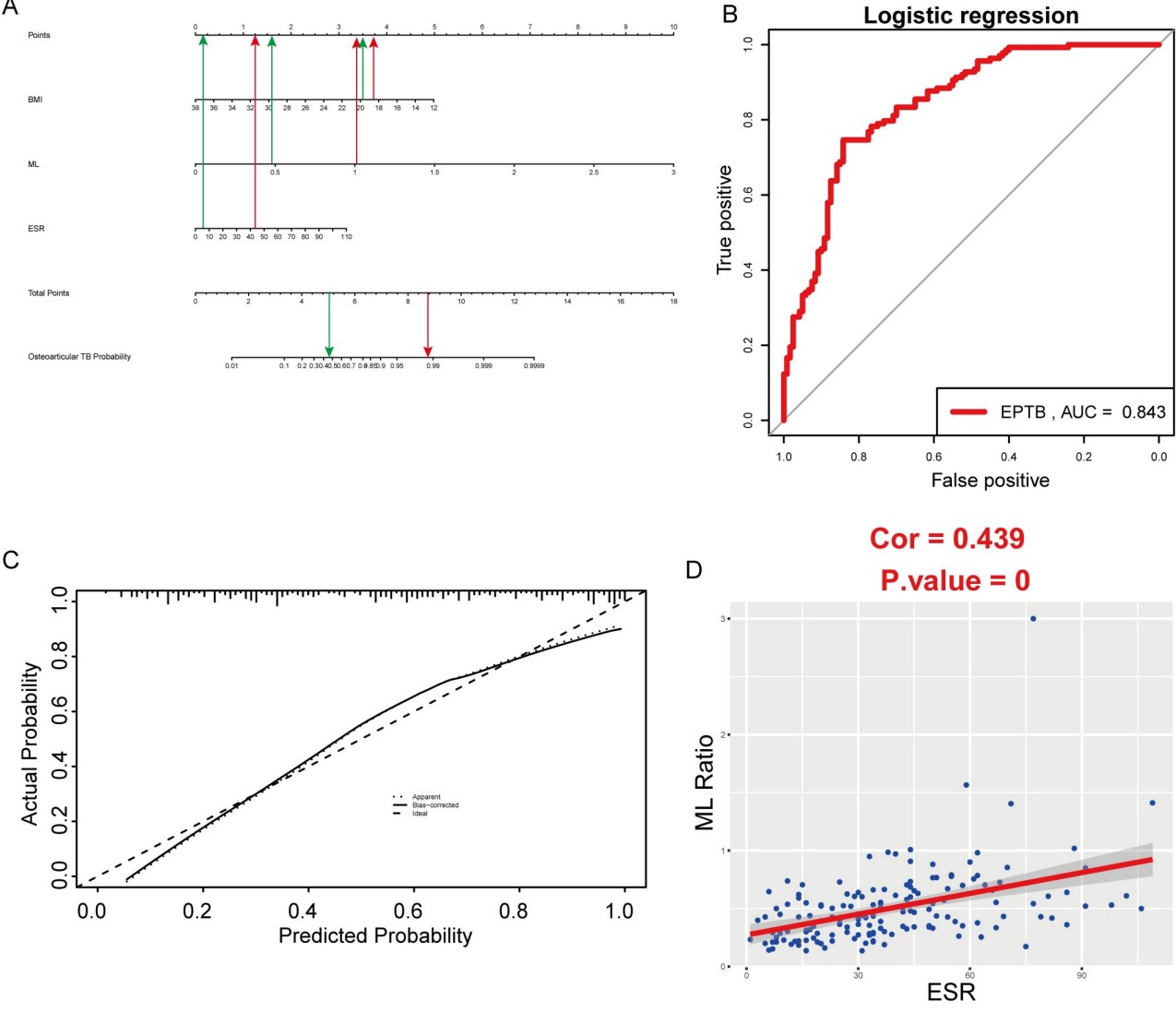

**Fig 3. Nomogram and correlation analysis.** (A) Nomogram for predicting osteoarticular TB probability for clinical data; The red line represents an osteoarticular TB patient while the green line represents an non-TB patient. (B) AUCs of the nomogram for clinical data. (C) Calibration curves for predicting osteoarticular TB probability for clinical data. (D) The correlation of ML ratio with ESR.

**Table 2. Univariate and multivariate logistic regression of the clinical data.**

| Characteristics | Univariate logistic | | Multivariate logistic | |
|---|---|---|---|---|
| | HR | P | HR | P |
| Age | 0.99 [0.98, 1.00] | 0.137 | | |
| Sex | 1.02 [0.66, 1.59] | 0.924 | | |
| BMI | 0.82 [0.75, 0.88] | p<0.0001 | 0.80 [0.72, 0.88] | p<0.0001 |
| White blood cell | 0.94 [0.87, 1.02] | 0.165 | | |
| Erythrocyte | 1.03 [0.92, 1.21] | 0.656 | | |
| ESR | 1.05 [1.03, 1.06] | p<0.0001 | 1.03 [1.01, 1.05] | 0.001 |
| Hemoglobin | 0.97 [0.96, 0.99] | p<0.0001 | 1.00 [0.98, 1.03] | 0.663 |
| ML ratio | 45.39 [13.37, 173.58] | p<0.0001 | 21.41 [2.42,222.93] | 0.008 |
| PL ratio | 1.01 [1.01, 1.01] | p<0.0001 | 1.00 [1.00, 1.01] | 0.1 |
| PM ratio | 1.00 [1.00, 1.00] | 0.384 | | |

just mentioned facts, we try to benefit from ML ratio, which can help us with osteoarticular TB diagnosis since it is difficult to obtain pus or tissue for *M.tb* culture [27]. Besides, a bacterial culture is frequently culture-negative. In this study, the ML ratio was an independent predictive factor for osteoarticular TB diagnosis in both the dataset GSE83456 and clinical datasets. Higher ML ratio was associated with osteoarticular TB through bioinformatic analysis and clinical validation. ESR is one of the blood markers monitor TB and infection [28]. Moreover, ESR is positively related to the ML ratio. As the results are shown in this studies, osteoarticular TB will lead to anemia, low albumin, and low BMI [29, 30]. Moreover, nutritional deficiencies predispose to a worse osteoarticular TB infection [31]. Results of the study indicate that the ML ratio, BMI, and ESR are independent predictive factors for osteoarticular TB diagnosis. Therefore, we constructed a nomogram to improve the diagnosis accuracy of osteoarticular TB. The AUC of the nomogram, which demonstrated more accurate and practical performance, is 0.843. Our study provides clear indicators, easily acquired in clinical work, for improvement of osteoarticular TB diagnosis. Nonetheless, the osteoarticular TB diagnosis also depends on the imaging. Combine our nomogram with imaging may significantly improve the diagnosis of osteoarticular TB. Therefore, our future research will focus on this combination.

There are three important potential limitations of this study: 1) since this is a retrospective study of osteoarticular TB and non-TB group, there is a possibility of selection bias; 2) The participants of this study did not have TSTs or interferon γ–release assays performed; 3) Most of the diagnosed patients were typically confirmed to be pathologically positive but microbiologically negative.

## Conclusion

We can conclude that the immune status of individuals changes during TB infection. Nonetheless, the ML ratio was identified as an independent predictive factor for EPTB diagnosis in the dataset GSE83456 and was considered a significant characteristic of osteoarticular TB. Lastly, the nomogram we constructed showed satisfactory ability to diagnose osteoarticular TB.

## Supporting information

**S1 Fig. Cell types in the microenvironment.** (A, B, C, D, E) xCell score of sixty-four cell types in GSE83456 were grouped into five groups: lymphocytes, myeloids, stem, stromal, and other cells.
(TIF)

**S2 Fig. Nomograms.** (A) Nomogram for predicting EPTB probability base on ML ratio in dataset GSE83456. (B) Nomogram for predicting osteoarticular TB probability base on ML ratio in clinical data.
(TIF)

**S1 File. xCell score of 64 subtypes immune cells of each sample.**
(XLSX)

## Acknowledgments

We are grateful to Dr. Xinli Zhan (Spine and Osteopathy Ward, The First Affiliated Hospital of Guangxi Medical University) for his kindly assistance in all stages of the present study.

## Author Contributions

**Conceptualization:** Tuo Liang, Xinli Zhan.

**Data curation:** Jiarui Chen, Zhaojie Qin, Zhen Ye.

**Formal analysis:** GuoYong Xu, Zide Zhang, Haopeng Zeng, Jie Jiang.

**Investigation:** Jiarui Chen, Zhaojie Qin, Zhen Ye.

**Methodology:** Tuo Liang, Xinli Zhan.

**Software:** GuoYong Xu, Zide Zhang, Haopeng Zeng, Jie Jiang.

**Visualization:** Jiang Xue, Tianyou Chen, Hao Li, Yunfeng Nie.

**Writing – review & editing:** Chong Liu.

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
