## [Decision Letter · Decision Letter 0]

28 Apr 2021

PONE-D-20-39926

Identification of Immune Status Changing in Patients with Bone Tuberculosis

PLOS ONE

Dear Dr. Zhan,

Thank you for submitting your manuscript to PLOS ONE. After careful consideration, we feel that it has merit but does not fully meet PLOS ONE’s publication criteria as it currently stands. Therefore, we invite you to submit a revised version of the manuscript that addresses the points raised during the review process.

I would like to sincerely apologise for the delay you have incurred with your submission. It has been exceptionally difficult to secure reviewers to evaluate your study. We have now received three completed reviews; their comments are available below.

Please revise the manuscript to address all the reviewer's comments in a point-by-point response in order to ensure it is meeting the journal's publication criteria. Please note that the revised manuscript will need to undergo further review, we thus cannot at this point anticipate the outcome of the evaluation process.

We look forward to receiving your revised manuscript.

Kind regards,

Miquel Vall-llosera Camps

Senior Editor

PLOS ONE

Journal Requirements:

2) We note that you have stated that you will provide repository information for your data at acceptance. Should your manuscript be accepted for publication, we will hold it until you provide the relevant accession numbers or DOIs necessary to access your data. If you wish to make changes to your Data Availability statement, please describe these changes in your cover letter and we will update your Data Availability statement to reflect the information you provide.

3) PLOS requires an ORCID iD for the corresponding author in Editorial Manager on papers submitted after December 6th, 2016. Please ensure that you have an ORCID iD and that it is validated in Editorial Manager. To do this, go to ‘Update my Information’ (in the upper left-hand corner of the main menu), and click on the Fetch/Validate link next to the ORCID field. This will take you to the ORCID site and allow you to create a new iD or authenticate a pre-existing iD in Editorial Manager. Please see the following video for instructions on linking an ORCID iD to your Editorial Manager account: https://www.youtube.com/watch?v=_xcclfuvtxQ

4) Please amend your list of authors on the manuscript to ensure that each author is linked to an affiliation. Authors’ affiliations should reflect the institution where the work was done (if authors moved subsequently, you can also list the new affiliation stating “current affiliation:….” as necessary).

5) Please include captions for your Supporting Information files at the end of your manuscript, and update any in-text citations to match accordingly. Please see our Supporting Information guidelines for more information: http://journals.plos.org/plosone/s/supporting-information.

Reviewers' comments:

Reviewer's Responses to Questions

**Comments to the Author**

1. Is the manuscript technically sound, and do the data support the conclusions?

Reviewer #1: Partly

Reviewer #2: No

Reviewer #3: Partly

2. Has the statistical analysis been performed appropriately and rigorously? 

Reviewer #1: Yes

Reviewer #2: I Don't Know

Reviewer #3: I Don't Know

3. Have the authors made all data underlying the findings in their manuscript fully available?

Reviewer #1: Yes

Reviewer #2: No

Reviewer #3: Yes

4. Is the manuscript presented in an intelligible fashion and written in standard English?

Reviewer #1: Yes

Reviewer #2: No

Reviewer #3: No

5. Review Comments to the Author

Reviewer #1: In this paper, the authors aim to identify economic and easily producible laboratory and clinical parameters, the variation of which can be associated with bone tuberculosis.

For this purpose, the authors first analyze a GSE83456 gene set to evaluate the different scores relating to the leukocyte populations in patients with bone tuberculosis and in healthy subjects. They then perform a similar analysis with the blood counts and clinical data of patients hospitalized for bone tuberculosis or for lumbar disc herniation or lumbar spinal stenosis.

Through a process of uni-variate and then multivariate logistic regression analysis, they first identify the ML ratio and then also the BMI and ESR as independent predictive parameters associated with the probability of bone tuberculosis. From these parameters they obtain more nomograms and test their validity through ROC curves.

The paper shows interesting results and potentially directly applicable to clinical practice; however, there are some points relating to the protocols applied and the results that require improvement before possible publication.

A limitation of this study, in my opinion, is that the group of NPTB was compared with only another group of non-infectious diseases. It would improve the results of the study, the comparison of the NPTB group with other experimental groups like LTBI or patients with non-tubercular infectious disease.

Major points

About the enrolled patients, it is not clear how the diagnosis of bone tuberculosis was made. The authors, in the material and methods section and table 1, should provide data relating to immunological tests, even if partial, or in any case, clarify how the diagnosis of bone tuberculosis was reached, e.g. by imaging.

In lines 27, 57, 71, etc. the authors refer to lymphoid cells and not lymphocytes, in supplementary table 1 there is no a lymphocyte population as a whole but 21 different populations classified as lymphoid; the authors should specify which of these populations were used for the calculation of the ML ratio obtained from the GSE83456 dataset and which differences, in terms of calculation, exist between this ML ratio and that obtained from the clinical data of the patients.

In Figure 2A it is not clear to me how the ML ratio was calculated from the GSE83456 dataset, is it a ratio between the scores?

In lines 105 and 106, the authors refer to healthy controls who actually are subjects suffering from disc herniation or lumbar spinal stenosis, for clarity it would be better to define them as non-TB patients.

In figure 1C the authors refer to a cell ratio, but it is not clear what this parameter indicates, is it perhaps the frequency of the single populations on the total of leukocytes?

In figure S1 there are two nomograms (A and B) but in the text, they are not described and it is not clear what difference there is between the two; the authors should clarify this point.

Minor points

In table 1, please correct the term eosinophil, in general table 1 should be better formatted.

Would be interesting if the authors, in fig 3A show where some NPTB and non TB patient lies in the nomogram.

The right citation for reference n15 is: La Manna MP, Orlando V, Dieli F, Di Carlo P, Cascio A, Cuzzi G, Palmieri F, Goletti D, Caccamo N. Quantitative and qualitative profiles of circulating monocytes may help identifying tuberculosis infection and disease stages. PLoS One. 2017 Feb 16;12(2):e0171358. doi: 10.1371/journal.pone.0171358. Among the authors, there is no Wilkinson KA.

Reviewer #2: Need to be re written. This manuscript does not reflect new findings . In addition there a wrong claim in it. The authors said that bone infection is the most common extrapulmonary Tb. This statement contradict most of the publications which said lymph node is the most dominant extrapulmonary type. The English language is very poor.

Reviewer #3: In this manuscript authors have studied immune status of extrapulmonary Tuberculosis patients, using a dataset of gene expression profiling of 47 EPTB and 61 healthy controls(GSE83456 ),which was downloaded from an open data source. The findings are compared with data collected from a hospital (study site) of 166 patient diagnosed having bone TB and 162 non-TB patients.

Furthermore, authors have constructed a nomogram based on three independent risk factors ML ratio, ESR, and BMI for clinical diagnosis of bone TB.

The manuscript requires a lot of English editing, and some sections need revision.

There are many ambiguities in method, result and discussion sections which need careful review and revision, given below are few examples.

Method: Type of demographic, clinical and laboratory data collected from TB and non-TB patient files (2012-2018) and source of cell counts used for comparison with GSE83456 data file

Results: Vague expressions and unspecific determinate words ["much lower”, “much higher” “satisfactory agreement”] are used that are not specific or precise enough for the reader to derive exact meaning.

Minor comments

For some statements, references are not cited (row 50-52 and 214-215)

Tables; Abbreviation used and unit of measurements are not described.

6. PLOS authors have the option to publish the peer review history of their article (what does this mean?). If published, this will include your full peer review and any attached files.

Reviewer #1: No

Reviewer #2: No

Reviewer #3: No

---

## [Author Response · Author response to Decision Letter 0]

17 May 2021

Dear Editors and Reviewers:

It is with excitement that I resubmit to you a revised version of the manuscript “Immune Status Changing Helps Diagnose Osteoarticular Tuberculosis” (ID: PONE-D-20-39926) for the “Plos One”. Thank you for giving me the opportunity to revise and resubmit this manuscript. I appreciate the time and detail provided by each reviewer and by you and have incorporated the suggested changes into the manuscript to the best of my ability. We highlight the changes to our manuscript within the document by using the track changes mode in Microsoft Word. The manuscript has certainly benefited from these insightful revision suggestions. I look forward to working with you and the reviewers to move this manuscript closer to publication in the “Plos One”. According to your nice suggestions, we have made extensive corrections to our previous draft. changes to the manuscript are shown in red. Point-by-point responses to the nice associate editor and three nice reviewers are listed below this letter.

Responses to the reviewer’s comments:

Reviewer #1: 

Major points

1. About the enrolled patients, it is not clear how the diagnosis of bone tuberculosis was made. The authors, in the material and methods section and table 1, should provide data relating to immunological tests, even if partial, or in any case, clarify how the diagnosis of bone tuberculosis was reached, e.g. by imaging.

Thank you for your valuable comments. We redefined the diagnostic and exclusion criteria for osteoarticular TB (Methods section, line 93-103, page 4).

2. In lines 27, 57, 71, etc. the authors refer to lymphoid cells and not lymphocytes, in supplementary table 1 there is no a lymphocyte population as a whole but 21 different populations classified as lymphoid; the authors should specify which of these populations were used for the calculation of the ML ratio obtained from the GSE83456 dataset and which differences, in terms of calculation, exist between this ML ratio and that obtained from the clinical data of the patients. 

Thank you for your valuable comments. Consistenting with the method of lymphocyte count in the routine blood examination, the xCell score of lymphocytes was a constitution of 21 subtypes (Supplemental Figure S1) of lymphoid cells (B cells, T cells and NK cells). We also point out that in the main text (Methods section, line 83-86, page 3). The ML ratio in the dataset GSE83456 represents the ratio of xCell score of monocytes to lymphocytes. The ML ratio in the clinical dataset represents the ratio of cell counts of monocytes to lymphocytes (Results section, line 152-154, page 6).

3. In Figure 2A it is not clear to me how the ML ratio was calculated from the GSE83456 dataset, is it a ratio between the scores?

Thank you for your valuable comments. The ML ratio in the dataset GSE83456 represents the ratio of xCell score of monocytes to lymphocytes. The ML ratio in the clinical dataset represents the ratio of cell counts of monocytes to lymphocytes (Results section, line 152-154, page 6).

4. In lines 105 and 106, the authors refer to healthy controls who actually are subjects suffering from disc herniation or lumbar spinal stenosis, for clarity it would be better to define them as non-TB patients.

Thank you for your valuable comments. In The First Affiliated Hospital of Guangxi Medical University, from 2012 and 2018, we randomly screened out the non-TB patients from all the inpatients diagnosed with lumbar disc herniation or lumbar spinal stenosis (Methods section, line 105-107, page 4).

5. In figure 1C the authors refer to a cell ratio, but it is not clear what this parameter indicates, is it perhaps the frequency of the single populations on the total of leukocytes? 

Thank you for your valuable comments. The cell ratio represents the frequency of the single populations on the total of leukocytes in the routine blood examination. Actually, these data had no effect on the results in this study. We have deleted figure 1C from figure 1 in this version of manuscript.

6. In figure S1 there are two nomograms (A and B) but in the text, they are not described and it is not clear what difference there is between the two; the authors should clarify this point.

Thank you for your valuable comments. We identified the ML ratio as an independent risk factor for the diagnosis of EPTB; therefore, nomograms (Supplemental Figure S2) based on the ML ratio were constructed for both dataset GSE83456 and clinical data (Results section, line 155-159, page 6). Figure S2A represents the nomogram for predicting EPTB probability base on ML ratio in dataset GSE83456. Figure S2B represents the nomogram for predicting osteoarticular TB probability base on ML ratio in clinical data (Figure legends, line 442-444, page 16).

Minor points

1. In table 1, please correct the term eosinophil, in general table 1 should be better formatted.

Thank you for your valuable comments. We have modified this part in the text (Table 1). 

2. Would be interesting if the authors, in fig 3A show where some NPTB and non TB patient lies in the nomogram.

Thank you for your valuable comments. We have modified this part in the text (Figure 3A). The red line represents an osteoarticular TB patient, while the green line represents a non-TB patient.

3. The right citation for reference n15 is: La Manna MP, Orlando V, Dieli F, Di Carlo P, Cascio A, Cuzzi G, Palmieri F, Goletti D, Caccamo N. Quantitative and qualitative profiles of circulating monocytes may help identifying tuberculosis infection and disease stages. PLoS One. 2017 Feb 16;12(2):e0171358. doi: 10.1371/journal.pone.0171358. Among the authors, there is no Wilkinson KA.

Thank you for your valuable comments. We have modified this part in the text as reference no.13 (Reference section, line 337-341, page 12). 

Reviewer #2

Need to be re written. This manuscript does not reflect new findings. In addition, there a wrong claim in it. The authors said that bone infection is the most common extrapulmonary Tb. This statement contradicts most of the publications which said lymph node is the most dominant extrapulmonary type. The English language is very poor.

Thank you for your valuable comments. Due to the lack of writing and English ability, the article before was poorly organized, so we search professional help for improvements to the English language within our manuscript.

Reviewer #3

1.The manuscript requires a lot of English editing, and some sections need revision. There are many ambiguities in method, result and discussion sections which need careful review and revision, given below are few examples.

Method: Type of demographic, clinical and laboratory data collected from TB and non-TB patient files (2012-2018) and source of cell counts used for comparison with GSE83456 data file.

Results: Vague expressions and unspecific determinate words ["much lower”, “much higher” “satisfactory agreement”] are used that are not specific or precise enough for the reader to derive exact meaning.

Thank you for your valuable comments. Due to the lack of writing and English ability, the article before was poorly organized, so we search professional help for improvements to the English language within our manuscript.

Minor comments

1. For some statements, references are not cited (row 50-52 and 214-215)

Thank you for your valuable comments. We have modified this part in the text. 

2. Tables; Abbreviation used and unit of measurements are not described.

Thank you for your valuable comments. We have modified this part in the text (Table 1). 

We tried our best to improve the manuscript and made some changes in the manuscript. These changes will not influence the content and framework of the paper. And here we did not list the changes but marked in red in revised paper. We appreciate for Editors/Reviewers’ warm work earnestly, and hope that the correction will meet with approval. 

Once again, thank you very much for your comments and suggestions.

---

## [Decision Letter · Decision Letter 1]

25 May 2021

Immune Status changing Helps Diagnose Osteoarticular Tuberculosis

PONE-D-20-39926R1

Dear Dr. Zhan,

We’re pleased to inform you that your manuscript has been judged scientifically suitable for publication and will be formally accepted for publication once it meets all outstanding technical requirements.

Kind regards,

Colin Johnson, Ph.D.

Academic Editor

PLOS ONE

Additional Editor Comments (optional):

Reviewers' comments:

Reviewer's Responses to Questions

**Comments to the Author**

1. If the authors have adequately addressed your comments raised in a previous round of review and you feel that this manuscript is now acceptable for publication, you may indicate that here to bypass the “Comments to the Author” section, enter your conflict of interest statement in the “Confidential to Editor” section, and submit your "Accept" recommendation.

Reviewer #1: All comments have been addressed

2. Is the manuscript technically sound, and do the data support the conclusions?

Reviewer #1: Yes

3. Has the statistical analysis been performed appropriately and rigorously? 

Reviewer #1: Yes

4. Have the authors made all data underlying the findings in their manuscript fully available?

Reviewer #1: Yes

5. Is the manuscript presented in an intelligible fashion and written in standard English?

Reviewer #1: Yes

6. Review Comments to the Author

Reviewer #1: The authors have checked and improved almost all the critical points that I had noticed, they have also improved the writing. In my opinion, the paper is now more clear and intelligible.

7. PLOS authors have the option to publish the peer review history of their article (what does this mean?). If published, this will include your full peer review and any attached files.

Reviewer #1: No

---

## [Editor Report · Acceptance letter]

7 Jun 2021

PONE-D-20-39926R1 

Immune Status changing Helps Diagnose Osteoarticular Tuberculosis 

Dear Dr. Zhan:

I'm pleased to inform you that your manuscript has been deemed suitable for publication in PLOS ONE. Congratulations! Your manuscript is now with our production department. 

Kind regards, 

on behalf of

Dr. Colin Johnson 

Academic Editor

PLOS ONE